# Contamination of 8.2 ka cold climate records by the Storegga tsunami in the Nordic Seas

Stein Bondevik[1] ✉, Bjørg Risebrobakken[2], Steven J. Gibbons [3],
Tine L. Rasmussen[4] & Finn Løvholt[3]

The 8200-year BP cooling event is reconstructed in part from sediments in the Norwegian and North Seas. Here we show that these sediments have been reworked by the Storegga tsunami – dated to the coldest decades of the 8.2 ka event. We simulate the maximum tsunami flow velocity to be 2–5 m/s on the shelf offshore western Norway and in the shallower North Sea, and up to about 1 m/s down to a water depth of 1000 m. We re-investigate sediment core MD95-2011 and found the cold-water foraminifera in the 8.2 ka layer to be re-deposited and 11,000 years of age. Oxygen isotopes of the recycled foraminifera might have led to an interpretation of a too large and dramatic climate cooling. Our simulations imply that large parts of the sea floor in the Norwegian and North Seas probably were reworked by currents during the Storegga tsunami.

Tsunamis can disturb the seabed and rework offshore sediments. Investigations after the 2011 Tohoku tsunami in Japan showed redeposited mud and sorted sand layers on the Sendai shelf down to ~100 m water depth[1]. Resuspended sediments were also carried offshore and continued down the slope to deep water, as turbidity currents[2]. The Storegga slide in the Norwegian Sea[3] triggered a giant tsunami[4]. Deposits from this tsunami have been extensively mapped onshore in Scotland e.g. refs. 5,6, Norway e.g. ref. 7, Shetland[6,8], and the Faroe Islands[9]. Currents in the Storegga tsunami must also have disturbed the seabed and reworked offshore sediments, but to what extent?

For long wave tsunamis, such as the one generated by the Storegga Slide, the currents are uniform from the surface to near the water/sediment interface at the sea floor and are a function of surface elevation (wave amplitude) and water depth. The maximum horizontal flow velocity, $u_{max}$, in a propagating tsunami can then be approximated to the maximum surface elevation $\eta_{max}$ through the expression:

$$u_{\max} = \frac{\eta_{\max}}{d} \cdot \sqrt{g \cdot d} \qquad (1)$$

where $d$ is water depth and $g$ is the gravitational constant. If the maximum sea surface elevation is 5 m and the water depth is 250 m, the maximum horizontal velocity will be approximately 1 m/s. According to the classical diagram of Hjulström[10], relating flow speed to erosion, transportation, and deposition of different grain sizes, a velocity of 1 m/s will erode and entrain a wide range of grain sizes, from fine silt to fine pebbles (5 μm to 6 mm).

The Storegga tsunami has been dated to the coldest decades of the 8.2 ka climatic event. Radiocarbon measurements of green moss fragments, picked out of the onshore tsunami deposits, were dated to $7300 \pm 20$ $^{14}$C years BP[11] and calibrated to 8080–8180 cal yr BP ($8140 \pm 55$ years ago, Supplementary Methods 1.1). The moss had still small amounts of intact chlorophyll, and thus must have been killed during the tsunami. From analysis of the growth pattern of these green moss stems it has been suggested that the tsunami happened in late autumn, possibly October–November[12]. According to the correlation with the Greenland ice cores, the date assigns the Storegga tsunami to the coldest part of the 160-year-long interval of the 8.2 ka event[11].

Here we ask the following question: *Could the 8.2 ka cold event, as reconstructed from marine sediment cores in the Norwegian and North*

[1]Department of Civil Engineering and Environmental Sciences, Western Norway University of Applied Sciences, P.O. Box 133, N-6851 Sogndal, Norway. [2]NORCE Climate & Environment, Bjerknes Centre for Climate Research, Bergen, Norway. [3]Norwegian Geotechnical Institute (NGI), P.O. Box. 3930 Ullevål Stadion, N-0806 Oslo, Norway. [4]Department of Geosciences, UiT the Arctic University of Norway, Tromsø, Norway. ✉e-mail: stein.bondevik@hvl.no

*Seas, be contaminated or confused with sediments re-deposited by the Storegga tsunami?* We answer by simulating the maximum flow velocity of the Storegga tsunami in the Norwegian Sea and North Sea and re-investigate a sediment core from the Vøring plateau, about 200 km offshore Norway in a water depth of 1050 m – one of the few North Atlantic marine sediment cores that recorded a distinct and dramatic cooling, the 8.2 ka event[13,14]. Our short answer to the question is yes.

## Results

### Computer simulations of the Storegga Slide and tsunami

We obtained flow velocities generated by the Storegga tsunami in the Norwegian and North Seas from computer simulations of the landslide and ensuing tsunami using a linear shallow water model e.g. refs. 4,15. The Storegga Slide was modeled as a cohesive clay-rich debris flow. The subsequent tsunami was simulated from the time-dependent changes in water depth caused by the moving landslide[16]. We chose parameters of the landslide rheology that best matched the run-out of

the landslide and the observed run-up heights of the tsunami deposits[17] (See methods for details).

The simulated maximum flow velocity representing the main part of the water column reflects the maximum wave amplitude and water depth. On the Norwegian shelf and in the North Sea, we calculate maximum velocities greater than 1 m/s at depths shallower than 250 m (Fig. 1, Supplementary Fig. 1). On the shallowest shelves, around Shetland, the Faroe Islands, and in Western Norway, maximum velocities are between 2 and 5 m/s. At many places with depth down to 1000 m we also simulated velocities above 1 m/s, especially in vicinity of the Storegga Slide (Fig. 1). The strongest simulated currents were due to the initial drawdown of water from the Norwegian shelf towards the backwall of the Storegga Slide; the currents here reached 5–10 m/s and the sea surface waters lowered by 15–30 m (Supplementary Fig. 1b). For the sites of interest, the simulated currents are uniform throughout the main part of the water column, with a thin boundary layer typically ranging from 1 to 10 m (see Methods). The simulated velocities shown in Fig. 1 apply almost down to the seafloor.

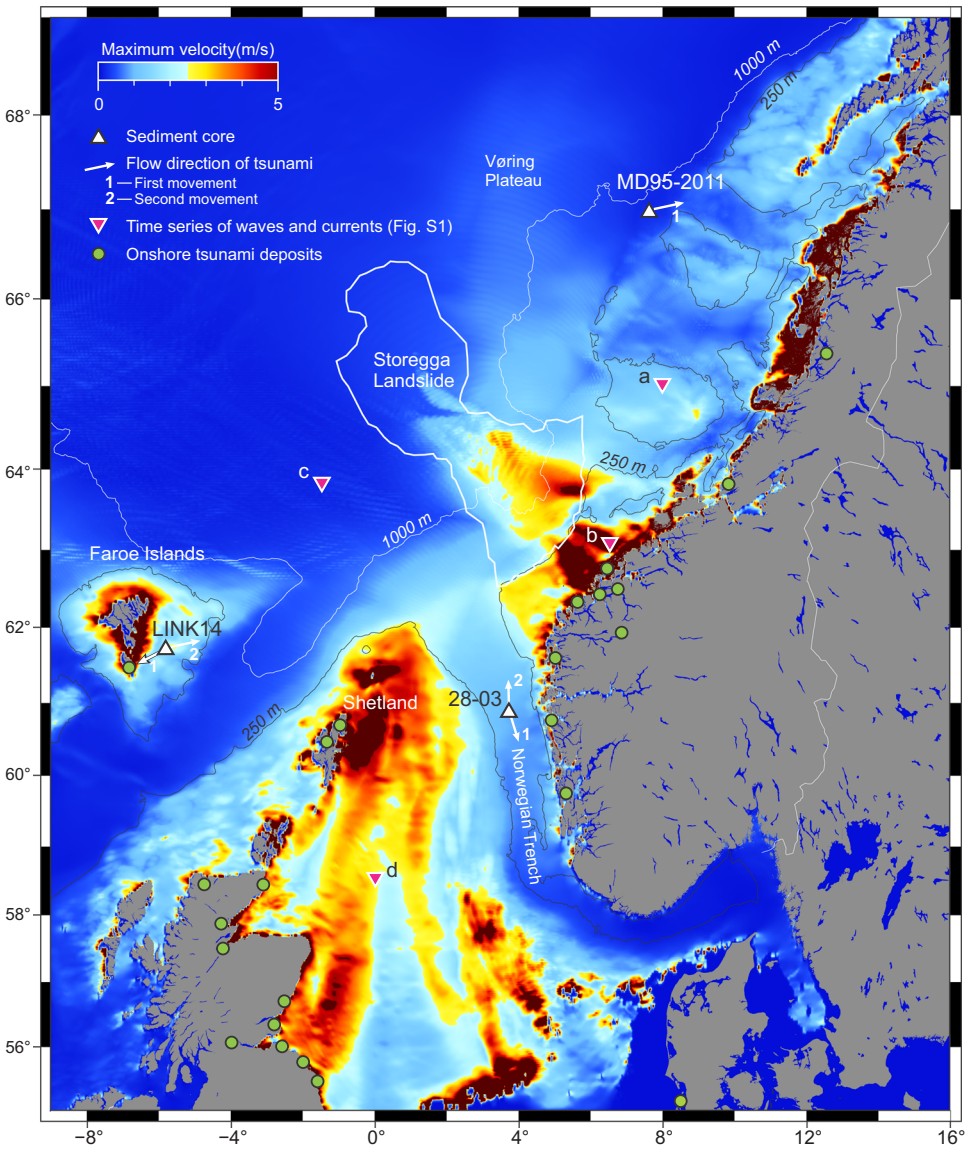

**Fig. 1 | Maximum flow velocity of the water column down to about 1–10 m above the sea floor during the Storegga tsunami.** Each pixel indicates the maximum flow velocity obtained during the 10 h simulation time. White triangles show location of marine sediment cores of the "8.2 event". Purple triangles (a–d) are additional locations of simulated time series (Supplementary Fig. 1). The white outline of the Storegga Slide is the run-out of landslide debris[34], the continuation of turbidites is not shown. Pixels in red-brown have maximum velocity >5 m/s and <20 m/s. Green circles are onshore locations of Storegga tsunami deposits[38]. Depth contours are paleo-bathymetry used in the simulations. Land is shown with present-day coastlines.

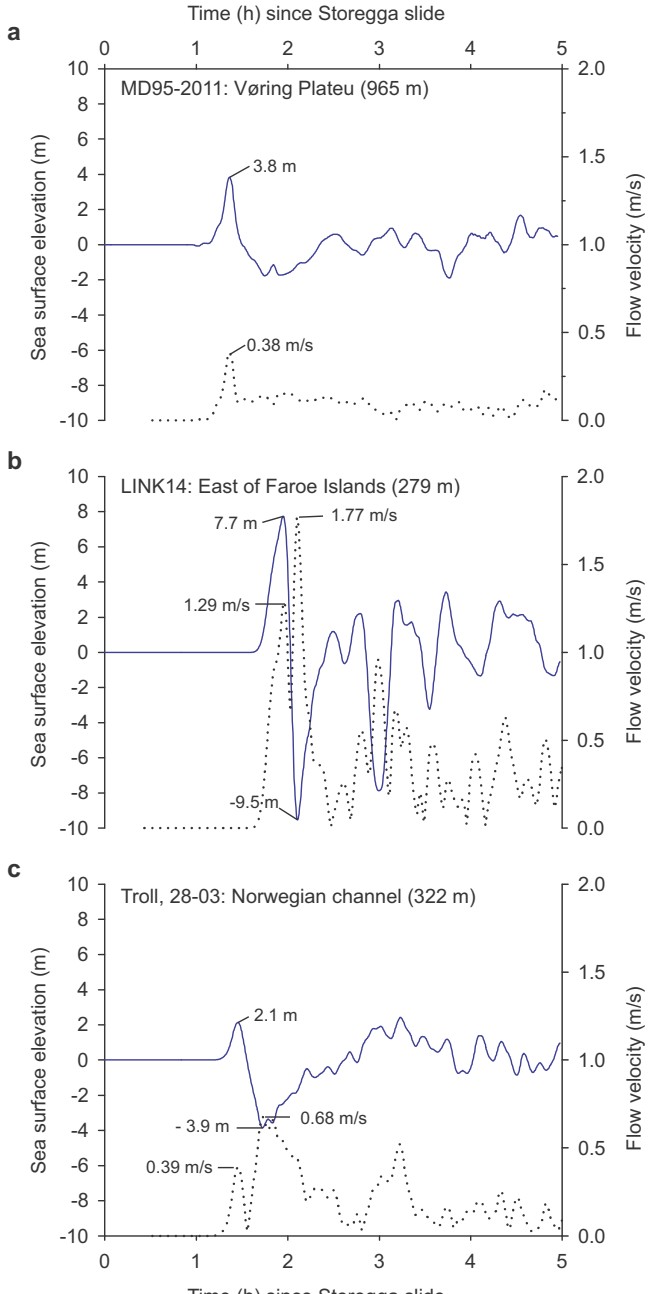

**Fig. 2 | Simulated tsunami surface elevation (blue line) and flow velocity down to about 1–10 m above the sea floor (dotted) during the Storegga tsunami at the location of marine sediment cores with an 8.2 ka layer.** Paleo-water depths in brackets. For core locations, see Fig. 1. **a** Core site MD95-2011 at the Vøring plateau in the Norwegian Sea. **b** Core site LINK14 east of the Faroe Island and **c** Core site Troll, 28-03 in the Norwegian channel. Source data are available at https://doi.org/10.18710/6KDQ7I.

## Re-investigation of the 8.2 ka cold event layer in the Norwegian Sea

We re-investigated sediment core MD95-2011 that had revealed very distinct changes identified as a response to the 8.2 ka climatic event[13,14]. The core was retrieved from the inner part of the Vøring plateau on the slope towards the shelf break (Fig. 1), in an area of high sedimentation rates[18]. Currents at the core location are at present too weak (<1 cm/s) to influence the bottom sediments[19]. However, during the Storegga tsunami, the simulations show a peak current speed of 39 cm/s corresponding to a wave amplitude of 3.8 m (Fig. 2a).

The proxies of the sediment core show a well-defined anomaly dated to around 8100 years ago. The anomaly, in a two-centimeter-thick layer, from 533–535 cm, shows a significant change in grain size, in the number of foraminifera, in the species of foraminifera, and in oxygen isotope values (Fig. 3).

The mixture of foraminiferal species in this layer indicates re-sedimentation. The abundances of both the cold polar species, *Neogloboquadrina pachyderma* and the warm Atlantic species *Neogloboquadrina incompta* increase 12–14 times (Fig. 3b). We also find a large number of benthic foraminifera that belong to environments of shallower water and stronger currents than presently found at the core site (Fig. 3c)[20]. Oxygen isotope values (δ[18]O) of *N. pachyderma* increase from 2.25 to 2.95‰ at this level (Fig. 3d) which would indicate a cooling of 3 °C of the surface ocean water. This anomaly is not seen in the oxygen isotopes of *N. incompta*[14].

We radiocarbon-dated the planktonic foraminifera in the 8.2 ka layer and found them to be much older than 8200 yr BP (Table 1). From the lowest centimeter of the layer (534–535 cm) we dated the cold-water species *N. pachyderma* to 10,690–11,170 cal yr BP and the warm-water species *N. incompta* to 8680–9190 cal yr BP (Table 1 and Fig. 3b). From the next cm up-core, within the layer, *N. pachyderma* was dated to 10,990–11,880 and *N. incompta* to 8670–9520 cal yr BP. All these ages are older than the radiocarbon dates below the layer, and especially the dates of *N. pachyderma* are about 2500–3000 years older than 8200 cal yr BP. The high δ[18]O value of this species (2.95‰) is comparable to the Early Holocene or Younger Dryas values in the same core[14] and in agreement with their radiocarbon ages.

The 8.2 ka layer fines upwards and the radiocarbon dates show a hiatus at the lower boundary that indicate erosion. The radiocarbon dates show a jump in ages across the layer and this jump indicates a time gap in the stratigraphy (Supplementary Fig. 4). We thus included a hiatus at the lower boundary (535 cm) in the age-depth modeling. The best fit shows a time gap or hiatus of 400 years (Supplementary Fig. 5). According to the sedimentation rates this corresponds to as much as 20 cm of erosion (stippled lines in Fig. 3a–d). The counts of mineral grains >150 μm, grain size and amounts of foraminifera, both planktonic and benthic (Fig. 3a, b), shows that the lowest centimeter (534–535 cm) is coarser grained than the next centimeter higher up (533–534 cm), indicating that the layer is normally graded.

All data support the hypothesis that the 8.2 ka layer in MD95-2011 is a turbidite. The erosive lower boundary, the upward fining of grains, the large amounts of redeposited older foraminifera and especially the high number of benthic foraminifera that thrive in shallower water confirm our suggestion that this layer is a fine grained turbidite deposited from the shelf break. Although the core site experienced currents directly from the propagating tsunami (Fig. 2a), we conclude that the 8.2 ka layer was deposited from a turbidity current released from sediments eroded and transported to the shelf break in the tsunami backwash.

## Discussion

Layers similar to the sand layer in MD95-2011 were found offshore Japan after the Tohoku tsunami in 2011. A fine-grained soft sediment layer covered wide areas of the deep sea outside the Sendai shelf after the Tohoku tsunami[1,21]. Evidence of the turbidity currents came from an ocean bottom pressure sensor, deployed at a water depth of 1052 m. The sensor was caught in the turbidity current and transported 1 km downslope three hours after the earthquake[2]. The interpretation is that the tsunami backflow carried sediments to the shelf and this cloud of suspended sediments rushed down the slope because of excess density. The moving suspension cloud grew into turbidity currents incorporating sea floor sediments and the ocean bottom pressure sensor. After 1 km downslope movement the 60 cm wide, 42-kilogram, sensor settled and, when the cloud came to rest, the sediments settled: sand first followed by finer grains. We think it is likely

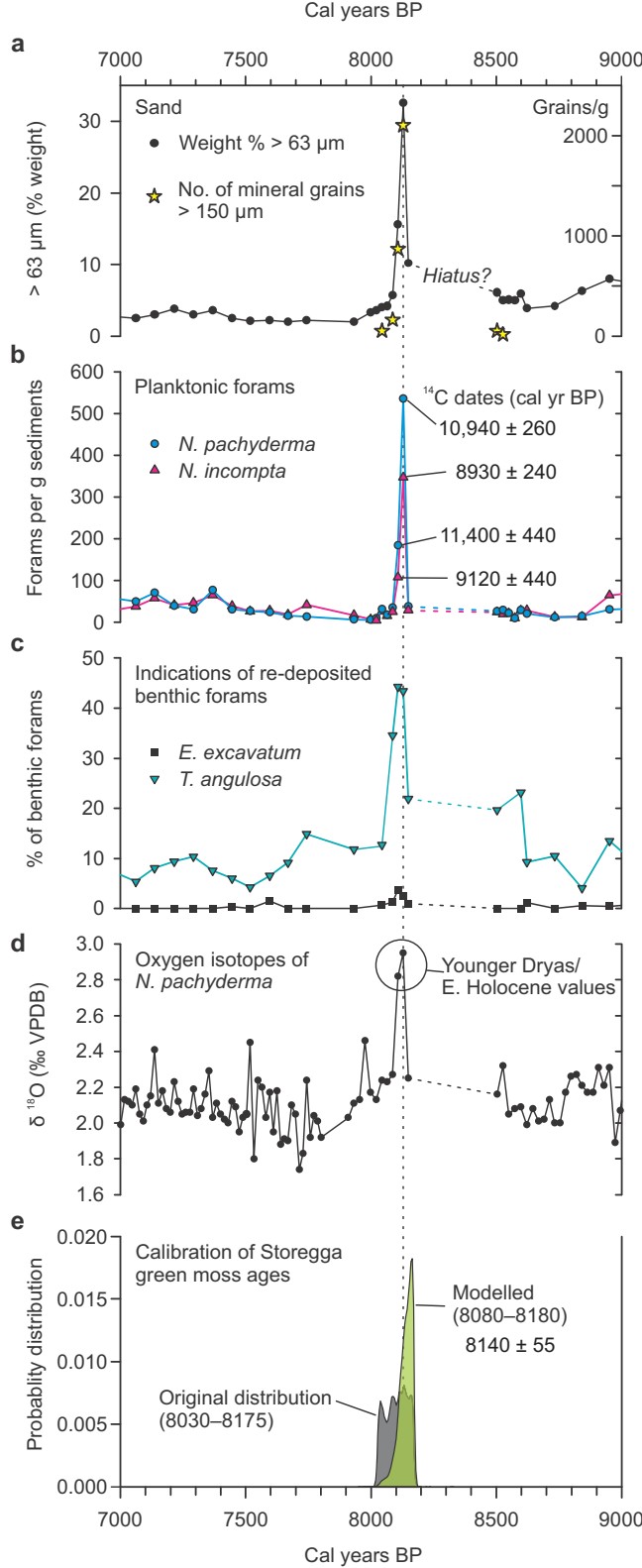

**Fig. 3 | Data from core MD95-2011 plotted according to age. a** Percentage of grains >63 µm and the number of grains >150 µm per gram sediments. **b** Number of planktonic foraminifera, the blue line is for the cold-water species *Neogloboquadrina pachyderma* and the red line is for the warm-water species *Neogloboquadrina incompta*. Calibrated [14]C dates are plotted (2 σ range). **c** Percentage of benthic foraminifera that indicate re-deposition, shelf species *Trifarina angulosa* and *Elphidium excavatum*[20]. **d** Oxygen isotopes of *Neogloboquadrina pachyderma*. **e** Green curve shows the modeled distribution of the ages of onshore green moss fragments within the Storegga tsunami, gray curve is the original distribution[11]. (See Supplementary Methods 1 and 2.1 for age modeling). Source data are available at https://doi.org/10.18710/6KDQ7I.

increase in larger-sized planktic and benthic foraminifera. Small planktic species are absent. The peculiar composition of the fauna, indicative of warmer sea surface temperatures, led the authors to conclude that this layer was subject to some kind of sorting effect, possibly from currents in the Storegga tsunami. The age–depth models of the radiocarbon ages (Supplementary Table 3) in the sediment core (Supplementary Figs. 6 and 7) suggest that the age of the layer is between 7930 and 8470 yr BP (2 σ range).

Our tsunami simulations show that the sediment core site of LINK14 would experience quite strong currents of up to about 1.8 m/s in the water column (Fig. 2b). Considering the boundary layer thickness, this current is roughly reduced to about 1.2 m/s one meter above the seafloor (Supplementary Table 2). These currents could hence transport sand from the shallower banks into the trough. Both the Sundborg diagram[23] and the equations by Miller et al.[24] show that such strong currents would erode sediments with grain sizes of fine silt to fine pebbles (Supplementary Table 2).

Core 28-03 in the Norwegian Channel (Fig. 1; Supplementary Table 1, Supplementary Methods 2.3) also show a distinct and sudden 8.2 ka event[25] that we think could be contaminated from currents in the Storegga tsunami. The layer assigned to the 8.2 ka event is 3–4 cm thick (-341–344 cm core depth), of clay-rich silts, and the age of the upper boundary is ca. 8200 cal yr BP (Supplementary Figs. 8 and 9). The strongest simulated currents are 0.5 m/s one meter above the seafloor (Fig. 2c) and are aligned parallel to the Norwegian Channel (Fig. 1). According to Supplementary Table 2 these currents could stir up fine-grained sediments (silt and sand) in the trench[26] and possibly rework and redeposit the foraminifera.

The simulations also reveal strong currents on the shelf around Iceland. Andrews and Giraudeau[27] found high amounts of reworked Neogene (Late Tertiary) coccoliths in a sediment core from a shallow basin at 164 m depth, dated to 8.2 ka BP. The peak of coccoliths was a unique signature of the 5.5 m long sediment core and was inferred to be the result of erosion of Neogene outcrops and re-sedimentation. Farther off the Icelandic coast, at 440 m water depth, there might also be indications of re-sedimentation in core MD99-2275 (Supplementary Figs. 2 and 3). Knudsen, et al. [28] found a distinct minimum in magnetic susceptibility followed by a peak of the epifaunal foraminifera *Cibicides lobatulus* between 8200 and 8000 cal yr BP. Radiocarbon ages of *C. lobatulus* are about 1500 years older than the molluscs[29]. The authors suggested that the peak in *C. lobatulus* might be due to reworking. We think it is likely that this erosion and reworking was caused by the Storegga tsunami.

Large parts of the ocean floor between Norway, Iceland, and Greenland could have experienced Storegga tsunami currents strong enough to rework and move sediments. According to our simulations, a maximum flow velocity of 25 cm/s or larger, capable of moving grains up to 1 mm, was simulated down to about 1000 m water depth between 58° and 74° N (Supplementary Fig. 3). In addition, turbidity currents could have been released from the shelf breaks forming turbidites – as demonstrated for core MD95-2011 at the Vøring plateau.

that similar sheet-like turbidity currents could have been initiated along the shelf areas in the Norwegian Sea due to suspension induced by the Storegga tsunami backwash.

Sediment core LINK14 from a trough on the eastern shelf of the Faroe Islands (Fig. 1, Supplementary Table 1, Supplementary Methods 2.2), has an '8.2 ka layer' interpreted by Rasmussen and Thomsen[22] as deposited directly from the Storegga tsunami currents. A fine sand layer (63–150 µm) between 114 and 116 cm core depth, shows an

**Table 1 | Radiocarbon measurements of foraminifera of the 8.2 ka layer in core MD95-2011**

| Lab.no | Depth (cm) | Species | Weight (mg) | δ¹³C (‰ PDB) | Radiocarbon yr BP | Cal. yr BP (2 σ-range)[a] | Mean cal. yr BP (2 σ-range) |
|---|---|---|---|---|---|---|---|
| Above the 8.2 ka layer | | | | | | | |
| ETH-130635 | 531–532 | *N. incompta* | 0.7 | –3.80 | 7810 ± 100 | 7980–8500 | 8240 ± 260 |
| ETH-130636 | 531–532 | *N.pachyderma* | 0.7 | –1.26 | 7600 ± 140[b] | 7710–8350 | 8030 ± 320 |
| Within the 8.2 ka layer | | | | | | | |
| ETH-130637 | 533–534 | *N. pachyderma* | 1.0 | –2.55 | 10220 ± 140 | 10,990–11,880 | 11,400 ± 440 |
| Tua-8053 | 534–535 | *N. pachyderma* | 8.3 | 1.0[c] | 9900 ± 60 | 10,690–11,170 | 10,940 ± 260 |
| Poz-8236 | 533–534 | *N. incompta* | 5.0 | | 8530 ± 160 | 8670–9520 | 9120 ± 440 |
| Tua-8054 | 534–535 | *N. incompta* | 8.1 | 1.0[c] | 8375 ± 55 | 8680–9190 | 8930 ± 240 |
| Below the 8.2 ka layer | | | | | | | |
| ETH-130638 | 536–537 | *N. incompta* | 0.7 | –2.88 | 7860 ± 110 | 8010–8580 | 8300 ± 280 |
| ETH-130639 | 536–537 | *N.pachyderma* | 0.67 | –2.32 | 8080 ± 100 | 8290–8900 | 8560 ± 300 |

[a]Radiocarbon ages were calibrated to calendar years with the dataset Marine20[39] using a local marine reservoir correction of ΔR = –145 ± 35 (Supplementary Methods 1.2).
[b]Age not used in the age modeling. Measured on the leached fraction of the foraminifera sample, main fraction too small to measure.
[c]Assumed value.

The previous climate reconstructions of a large and abrupt 8.2 ka cooling from marine sediment cores in this area are thus faulty and should be discarded. Instead, sea floor sediments showing a strong 8.2 ka anomaly with evidence of re-working should rather be considered as a deep-water Storegga tsunami deposit.

## Methods

### Computer simulation of Storegga slide and tsunami

The tsunami generated by the Storegga slide is modeled in two stages. First, the dynamics and time-evolution of the slide itself are simulated using the two-layer depth-averaged BingCLAW model[17], developed for cohesive clay-rich landslides. Secondly, the tsunami propagation, driven by the time-dependent changes in the water depth modeled by BingCLAW, is simulated using the GloBouss tsunami model e.g[30–32]. The simulations used ocean depths as derived by Hill et al. [33]. that account for changes in bathymetry since 8150 yr BP, covering a region from 12.5°W to 16.6°E and 53.3°N to 70.0°N on a grid with approximately 2 km spacing. Both landslide and tsunami simulations were run for a duration of 10 h which was deemed sufficient to capture the evolution of the water wave over all geographical regions of interest.

The landslide dynamics and runout are controlled by several parameters describing the rheology of the flow. We chose the landslide parameters that gave the best match to the observed tsunami run-up heights[4], as guided by the sensitivity study of Kim, et al. [17]. The landslide simulation in the current study uses a volume of 3200 km³, an initial yield strength, $\tau_i$, of 12 kPa, a residual yield strength, $\tau_r$, of 3 kPa, a remolding coefficient, $\Gamma$, of 0.0005, and an added mass parameter, $c_m$, of 0.1. All other parameters are held to the fixed values employed by Kim et al. [17]

In this study we have used a landslide volume of 3200 km³, close to the original volume estimate[34]. Recently Karstens et al. [35] suggested that the volume of the Storegga slide should be reduced to 1300–2300 km³. We emphasize that such a reduced volume is not expected to change the conclusions found herein for the following reasons: Firstly, the simulated waves and currents are calibrated towards observed tsunami run-up heights e.g.[4]. The landslide is strongly linked to the properties of the slide material at the time, which are uncertain[17]. By reducing the strength properties of the slide material, we can likely reproduce the same amplitudes and currents of the tsunami with such a smaller landslide volume. Secondly, the wave generation is dominated by the slide material from farthest up the slope, in shallow water. In the study by Karstens et al. [35] the location of the slide material is not much altered compared to the original reconstruction.

### Estimating critical velocity for erosion

Flow velocities in a tsunami are mainly uniform throughout the water column apart from in a boundary layer near the seafloor. In deep water the boundary layer for a tsunami is relatively thin compared with the water depth. For the Storegga Slide tsunami the boundary layer will typically be about 1–10 m thick[36], and will vary with the current velocity, wave period, and surface roughness.

To estimate the critical velocity for erosion at the three sediment core sites LINK14, 28-03, and MD95-2011 (Fig. 1, Supplementary Table 1) we first consider the boundary layer thickness influenced by bottom friction for each site using the results from the simulations by Williams and Fuhrman[36]. From visual inspection of those simulations, we found that the current speed 1 m above the sea floor was reduced by a factor of about 0.7 (core site LINK 14) to 0.95 (core site MD95-2011) of the uniform water speed in the water column. Next, we found the largest and smallest erodible sediment grain sizes by reading off the grain sizes in the Sundborg diagram[23] for each current velocity and by calculations using the equations in Miller et al.[24]; see Supplementary Table 2 for details. The methods provide similar results, and both estimated higher velocities than needed to erode the grain sizes in the 8.2 ka layer at the sediment core sites LINK14 and 28-03.

## Data availability

The data presented are included in this published article and in the supplementary information file. Outline of the Storegga Slide is available from ref. 34. The paleo-bathymetry used in Fig. 1 is from ref. 33 and we are grateful to J. Hill for providing us with this. The bathymetry data for the wider calculations (Supplementary Figs. 2 and 3) is from https://www.gebco.net/data_and_products/gridded_bathymetry_data/. Source data for Figs. 2 and 3 are available at https://doi.org/10.18710/6KDQ7I.

## Code availability

The source code for the BingCLAW program that computes the evolution of the viscoplastic debris flow is available from https://github.com/norwegian-geotechnical-institute/BingCLAW_5.6.1 (last accessed 2023/12/06). BingCLAW requires the CLAWPACK software library and all its dependencies. CLAWPACK is available for download from http://www.clawpack.org/ (version 5.6.1 required: last accessed 2023/12/06). The source code to the GloBouss software for calculating the tsunami is found on https://github.com/geirkp/geirkp.github.io (last accessed 2023/12/06). Details of the implementation of the numerical model are found in the document https://github.com/geirkp/geirkp.github.io/

blob/master/SUP/globouss/rapport.pdf. The maps (Fig. 1, Supplementary Figs. 2 and 3) of the output from the tsunami simulation is generated using GMT software[37] available from https://www.generic-mapping-tools.org/ (last accessed 2023/12/06).

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

## Acknowledgements

Morten Hald suggested us to radiocarbon date the foraminifera at the 8.2 ka spike. Maarten Blaauw answered questions about the age-depth modeling. We are further indebted to David Furhman for discussions related to interpreting the tsunami boundary layers. Radiocarbon measurements were supported by Research council of Norway, project no. 325333 (B.R.). Adaptations to the tsunami model were performed under the "LAST - Life After the Storegga Tsunami" project funded by the Norwegian Research Council, project number 302858 (S.G. and F.L.).

We are grateful to Jon Hill for providing a file of the paleo-bathymetry from his 2014 paper on which the tsunami simulation was calculated.

## Author contributions

S.B. conceived the idea of the paper and wrote the first draft of the manuscript. B.R. led the re-investigation of sediment core MD95-2011 and picked foraminifera for radiocarbon dating. S.G. simulated the currents and surface elevation of the Storegga tsunami under the guidance of F.L. F.L. carried out the erosion assessment. T.L.R. picked foraminifera for radiocarbon dates in sediment core LINK14. S.B., B.R., S.G., T.L.R. and F.L. contributed to the final version of the paper.

## Funding

## Competing interests

The authors declare no competing interests.
