## [Peer Review File · Nature Communications]

Contamination of 8.2 ka cold climate records by the Storegga tsunami in the Nordic SeasREVIEWER COMMENTS

Reviewer #1 (Remarks to the Author):

This is a well-documented paper which casts doubt on the credibility of records purporting to show the 8.2 ka event in cores from the Nordic Seas. The data are of high quality, well set out in figures and tables, and entirely credible. It is clear that for evidence of the 8.2 ka event cores from other locations must be employed. The Storegga slide caused a well-documented tsunami for which bottom velocities are calculated here and demonstrated to be adequate to resuspend shallow sediments which then flow down slope in the form of turbidity currents contaminating any climatic records of that time. This is a significant and important result as negative data is not always accorded the significance that is due to it.

I do not think the words 'Confusion and' are needed in the title as 'contamination' is quite sufficient. One can reasonably conclude that one will lead to the other.

In places I find the authors to be rather too tentative, for example on line 44 'most likely' could be removed, also 'could' on line 167, and 'likely' rather than 'possible' on lines 203 and 240.

There are two aspects that need revision by the authors.

1. They are using three terms for the water surface perturbation due to the Storegga slide; 'surface elevation', 'wave amplitude', and 'wave height'. In figure 2 the caption is 'wave height' but the reference to it on line 115 is 'wave amplitude'. Normally in wave theory wave height is twice the wave amplitude. The authors need to sort out the terminology here. Is 'surface elevation' wave height?
2. The authors should not be using the archaic Hjulstrom curve for critical erosion conditions. This was improved by his student Sundborg (1956), but the most commonly used curve based on analysis of large amounts of data is that due to Miller et al (1977). These authors give an equation for the flow velocity measured at 1 m above the bed for the coarse end of the grain size spectrum ($D > 2$ mm) as $U = 160D^{0.45}$ giving a speed range of 0.6 to 1.2 m/s for 1 to 5 mm coarse sand-fine gravel. This, as with the condition based on Yalin (1977) which they cite, is also based on Shields curve. The period of the tsunami wave is sufficiently long (>20s) that the flow may be considered unidirectional and thus the wave velocity of Komar & Miller (1973, 1975) is not needed.

References.

Komar, P.D & M.C. Miller, 1973. The threshold of sediment movement under oscillatory water waves. *Journal of Sedimentary Petrology*, 43, 1101-1110.

Komar, P.D & M.C. Miller, 1975. Comparison between threshold of sediment motion under waves and unidirectional currents with a discussion of practical evaluation of threshold. *Journal of Sedimentary Petrology*, 43, 362-367

Sundborg, A, 1956. The River Klarälven; a study in fluvial processes. *Geografiska Annaler*, 38, 125-316.

Miller, M.C., McCave, I.N. and Komar, P.D. 1977. Threshold of sediment motion under unidirectional currents. *Sedimentology*, 24, 507-527.

Reviewer #2 (Remarks to the Author):

The manuscript by Bondevik et al combines numerical modeling and sediment core records to argue the sediment core records had been more likely contaminated and falsely used to refer the 8.2ka cold climate due to the rework of sediments by the Storegga landslide tsunami. In that regard, this manuscript provides rational evidences after re-examining offshore and nearshore cores, which are further explained using model simulation of the Storegga tsunami, specifically using the tsunami-induced current speed. The methods used in this study are technically correct, and they favorably support the quality of the data obtained with appropriate techniques. The conclusions arrived from this study may generate potential impact on how the 8.2ka cold climate event was defined and what magnitude of impact it had posed regionally and globally. The conclusion that "a strong 8.2 ka anomaly with evidence of re-working should rather be considered as a deep-water Storegga tsunami deposit" may be overstated as there have been evidences other than the sediments supporting the 8.2

ka code event. However, I admit that interpretation of geological core materials relating to specific causes is outside the scope of my expertise, and I am unable to assess fully. My comments below are mostly focused on numerical modeling of the tsunami and sediment transport.

Using numerical models, the manuscript carefully investigated the current speed generated by the Storegga tsunami at a few sediment cores previously obtained offshore and nearshore. The initiation and movement of the sediments are quantitatively explained using the model results. The manuscript clearly shows the sediments indicating the 8.2 ka cold climate had been reworked during the Storegga tsunami event. However, the sediment deposition and erosion at these cores are not clearly discussed and sufficiently illustrated, and can be improved from the following aspects:

- The time series of wave height and flow speed (not velocity) in Fig. 2 and Supplementary Fig. 1 clearly shows significant drawdown of water either as the leading wave (the East of Storegga backscarp) or following the first wave crest, and the largest flow speeds are mostly associated with these receding waves (certainly a normal phenomenon during a landslide tsunami). Wouldn't these receding waves result in more erosion than deposition, particularly in the nearshore area, and as a result affect how one interprets the causes of the sediment layers indicating the 8.2 Ka event? What roles do these receding flows play in correlating the sediment core records to the 8.2 ka event? That said, it will be helpful to include vectorized flow velocities in those time series plots and offer more insightful discussions.
- The manuscript offers only very brief discussions on "Estimating critical velocity for erosions". The comparisons between the modeled velocity and the critical velocities could have been provided in a table in the supplement. I would also strongly suggest the authors to take a more rigorous step: conduct model simulation on sediment transport (assuming such a modeling tool is available to authors) to approximate the amounts and deposition and erosion distributions of the sediments for the Storegga tsunami. This additional modeling is expected to clarify the sedimentation process at the core sites, and endow more solid and more complete conclusions. For this reason, I consider this is a lack of sufficient detail in the present manuscript to fully support their conclusions, a standard required by Nature Communications.
- One more detail the manuscript can provide is how the bottom friction was taken into account in the model simulation, and how much it affects the flow velocities.

Based on my comments above, my overall recommendation is a minor-major revision depending how much more work the authors are willing to invest in this study.

Response to the review of our manuscript:

Confusion and contamination of 8.2 ka cold climate records caused by the Storegga tsunami in the Nordic Seas

[Text in blue is our response, text in black is the reviewers]

We thank you for the review and the opportunity to resubmit the paper for possible publication in Nature Communications! Both reviewers are happy with the documentation and evidence we provide that the record of the well-known 8.2 ka cold climate event, as inferred from marine sediment cores in the Norwegian and Nordic Seas, is contaminated and disturbed by the Storegga tsunami. This is the main point of our paper. We are very pleased that the reviewers find our paper “well-documented and of high quality, well set out in figures and tables, and entirely credible” (reviewer 1).

The reviewers and editor ask us to provide more discussion and modeling of sediment deposition and erosion. We have revised our approach and provide a modification to the modeling based on the request from both reviewers. To this end, we have used the sediment model as suggested by reviewer #1. In addition, we have investigated the effect of the boundary layer to the tsunami current by consulting results from sophisticated modelling approaches considering the local shear profile at the sea floor that influences the erosion (motivated by comments from reviewer #2). The revised approach clearly improves the modelling, and we are hence thankful for the reviewer’s suggestions. It further consolidates the findings already reported in the first draft of this paper, with only slightly revised values for the erodible grain sizes.

Hence, the revision we have carried out, based on the reviewer’s comments is important because it clearly shows the potential for re-sedimentation and movement of sediment grains. On the other hand, it is beyond the scope of this paper to model in detail the erosion and sedimentation pattern, this should and must be left for further studies. In addition, we stress that erosion potential is largely based on empirically based thresholds, and we point clearly to current velocities that exceed these thresholds in this paper.

Below we address point by point the comments and critique raised by the reviewers.

Reviewer 1

I do not think the words ‘Confusion and’ are needed in the title as ‘contamination’ is quite sufficient. One can reasonably conclude that one will lead to the other.

We agree and have now deleted the word ‘Confusion’ from the title. The title is changed to:

“Contamination of 8.2 ka cold climate records caused by the Storegga tsunami in the Nordic Seas”

In places I find the authors to be rather too tentative, for example on line 44 ‘most likely’ could be removed, also ‘could’ on line 167, and ‘likely’ rather than ‘possible’ on lines 203 and 240.

There are two aspects that need revision by the authors.

Thanks for pointing this out. We have changed the text as suggested.

1. They are using three terms for the water surface perturbation due to the Storegga slide; 'surface elevation', 'wave amplitude', and 'wave height'. In figure 2 the caption is 'wave height' but the reference to it on line 115 is 'wave amplitude'. Normally in wave theory wave height is twice the wave amplitude. The authors need to sort out the terminology here. Is 'surface elevation' wave height ?

In the caption to Fig. 2 we have changed wave height to sea surface elevation.

In line 115 we use the word "amplitude" because it is the amplitude of the wave we are referring to. Here is the sentence: *However, during the Storegga tsunami the simulations show a current speed of 39 cm/s corresponding to a wave amplitude of 3.8 m (Fig. 2a)*. In Fig. 2a the amplitude is 3.8 m – it is the distance from the zero-line to the wave top. We could also use "surface elevation" here, but we think that using the word "wave amplitude" we indicate to the reader that it is the top point of the first wave we address, instead of saying "surface elevation".

Is 'surface elevation' wave height?

No. The sea surface elevation is the height of the sea surface at any given time. It is not the same as wave height or wave amplitude. We have taken care to make sure that all occurrences of the terms are now the correct ones.

2. The authors should not be using the archaic Hjulstrom curve for critical erosion conditions. This was improved by his student Sundborg (1956), but the most commonly used curve based on analysis of large amounts of data is that due to Miller et al (1977). These authors give an equation for the flow velocity measured at 1 m above the bed for the coarse end of the grainsize spectrum ($D > 2$ mm) as $U = 160D^{0.45}$ giving a speed range of 0.6 to 1.2 m/s for 1 to 5 mm coarse sand-fine gravel. This, as with the condition based on Yalin (1977) which they cite, is also based on Shields curve. The period of the tsunami wave is sufficiently long (>20s) that the flow may be considered unidirectional and thus the wave velocity of Komar & Miller (1973, 1975) is not needed.

We thank the reviewer for this comment. Based on the suggestion, we have adopted the curve by Miller et al. (1977). This has further been combined with an estimate of the boundary layer that reduces the velocity at 1 m height by a factor of 0.7-0.95 of the uniform water velocity. We have described this in the first paragraphs of the method section and also added a table with details in the supplements (Supplementary Table 2). However, this much improved method only slightly modifies the results and reaffirms the conclusions from the first version of the manuscript.

Reviewer #2 (Remarks to the Author):

The manuscript by Bondevik et al combines numerical modeling and sediment core records to argue the sediment core records had been more likely contaminated and falsely used to refer the 8.2ka cold

climate due to the rework of sediments by the Storegga landslide tsunami. In that regard, this manuscript provides rational evidences after re-examining offshore and nearshore cores, which are further explained using model simulation of the Storegga tsunami, specifically using the tsunami-induced current speed. The methods used in this study are technically correct, and they favorably support the quality of the data obtained with appropriate techniques. The conclusions arrived from this study may generate potential impact on how the 8.2ka cold climate event was defined and what magnitude of impact it had posed regionally and globally.

The conclusion that “a strong 8.2 ka anomaly with evidence of re-working should rather be considered as a deep-water Storegga tsunami deposit” may be overstated as there have been evidences other than the sediments supporting the 8.2 ka code event. However, I admit that interpretation of geological core materials relating to specific causes is outside the scope of my expertise, and I am unable to assess fully. My comments below are mostly focused on numerical modeling of the tsunami and sediment transport.

We thank the reviewer for the comments to our manuscript. Partly based on the comments below, and partly based on reviewer #1 comments, we have slightly revised our approach to assess the erosion conditions on the sea floor. In the revised paper, we include other empirical sediment erosion models, and also more carefully evaluate the flow field above the seafloor. Yet, the conclusions from the first version of the paper are still retained with the new approach (see above comments to reviewer #1).

The conclusion that “a strong 8.2 ka anomaly with evidence of re-working should rather be considered as a deep-water Storegga tsunami deposit” may be overstated as there have been evidences other than the sediments supporting the 8.2 ka cold event.

We do not say that the 8.2 ka cold event does not exist! Our finding and conclusion are that what has been taken as the 8.2 cold climate event deposits in the Norwegian and North Seas are rather deposits from the Storegga tsunami.

Using numerical models, the manuscript carefully investigated the current speed generated by the Storegga tsunami at a few sediment cores previously obtained offshore and nearshore. The initiation and movement of the sediments are quantitatively explained using the model results. The manuscript clearly shows the sediments indicating the 8.2 ka cold climate had been reworked during the Storegga tsunami event. However, the sediment deposition and erosion at these cores are not clearly discussed and sufficiently illustrated, and can be improved from the following aspects:

- The time series of wave height and flow speed (not velocity) in Fig. 2 and Supplementary Fig. 1 clearly shows significant drawdown of water either as the leading wave (the East of Storegga backscarp) or following the first wave crest, and the largest flow speeds are mostly associated with these receding waves (certainly a normal phenomenon during a landslide tsunami). Wouldn't these receding waves result in more erosion than deposition, particularly in the nearshore area, and as a result affect how one interprets the causes of the sediment layers indicating the 8.2 Ka event? What roles do these receding flows play in correlating the sediment core records to the 8.2 ka event? That said, it will be helpful to include vectorized flow velocities in those time series plots and offer more insightful discussions.

It is not the scope of this article to conduct a time dependent analysis of the detailed patterns of erosion and deposition of the Storegga tsunami. Rather, we attempt to analyze whether the current speed has the potential to erode the sea floor. This is done by comparing the modeled wave current

with state-of-the-art empirical models for erosion, and our analysis shows clearly that the current speeds exceed the erosion potential. We agree that this suggestion would be an interesting analysis. It would however add significant additional complications and it is not the target of this paper.

- The manuscript offers only very brief discussions on “Estimating critical velocity for erosions”. The comparisons between the modeled velocity and the critical velocities could have been provided in a table in the supplement.

Yes, good idea, we have added such a table in the supplement, Supplementary Table 2.

I would also strongly suggest the authors to take a more rigorous step: conduct model simulation on sediment transport (assuming such a modeling tool is available to authors) to approximate the amounts and deposition and erosion distributions of the sediments for the Storegga tsunami. This additional modeling is expected to clarify the sedimentation process at the core sites, and endow more solid and more complete conclusions. For this reason, I consider this is a lack of sufficient detail in the present manuscript to fully support their conclusions, a standard required by Nature Communications.

As stressed above, time dependent sediment transport modelling is not the main scope here. The main purpose is to assess whether the current velocities have substantial erosion potential at the sediment core sites. On the other hand, we have revised our modelling approach conducting the following two steps: Based on reviewer #1's comment, we have included the erosion models of Miller et al. (1977). As a second step, we have analyzed the effect of the boundary layer thickness for estimating the velocity at the 1 m height above the seafloor needed as input to the erosion models. In our opinion this is the most important modification as we now more carefully evaluate the flow field right above the seabed. We find that the speed used for estimating the erosion (one meter above the seafloor) is reduced by factors of 0.7-0.95 from the current speed shown in Fig. 2. This only slightly modifies the results and reaffirms the conclusions from the first version of the manuscript.

- One more detail the manuscript can provide is how the bottom friction was taken into account in the model simulation, and how much it affects the flow velocities.

As this is deep-water tsunami propagation, the friction force on the wave is of minor importance and it is hence omitted in the simulations.

Based on my comments above, my overall recommendation is a minor-major revision depending how much more work the authors are willing to invest in this study.

REVIEWER COMMENTS

Reviewer #1 (Remarks to the Author):

Review of the resubmission of "contamination of 8.2 K a cold climate records caused by the Storegga tsunami in the Nordic seas", by S. Bondevik et al.

Overall the authors have responded satisfactorily to (both) reviewers comments. The authors make a strong case for not interpreting data from the Nordic seas and North Sea around the time of the 8.2 event as evidence for that event. There are plenty of cores from outside this region which demonstrate the impact of the event on the marine realm.

There are only minor matters to highlight.

lines

31. ... 'might have been' , better 'probably were'.

55-8 and 223-4. It is not clear why the authors persist in citing the antique diagram of Hjulstrom (other than Nordic solidarity) which is incorrect in both its designation of the deposition field and in plotting the erosion resistance of fine sediment as a function of grainsize - see work of Dade et al. Post-war (WW II) and modern treatments of critical erosion do not mention Hjulstrom, for example the books of Yalin, Raudkivi, Mehta and Nielsen.

88. (102) I think the authors need to specify where their maximum flow velocity occurs, either at the seafloor or is it at the top of the boundary layer (the latter).

155-7. pachyderma

369, if they persist, needs date.

397, 413, 423, capitalisation,

References.

Dade, W.B., Nowell, A.R.M. and Jumars, P.A., 1992. Predicting erosion resistance of muds. *Mar. Geol.*, 105, 285-297.

Dade, W.B., and Nowell, A.R.M., 1991, Moving muds in the marine environment, In *Proceedings Coastal Sediments '91*, Water Resources Division Am. Soc. Civil. Eng., pp. 54-71.

Mehta, A.J., 2023. *An introduction to hydraulics of fine sediment transport*. Second edition, World scientific, Singapore, 1027 pp.

Nielsen, P., 1992. *Coastal bottom boundary layers and sediment transport*. World scientific, Singapore, 324 pp.

Raudkivi, A.J., 1976. *Loose boundary hydraulics*. 2nd edition, Pergamon press Oxford, 397 pp.

Yalin, M. S., 1977, *Mechanics of Sediment Transport*, 2nd Edition, Pergamon Press.

Reviewer #2 (Remarks to the Author):

The revised manuscript is a much-improved version from the original. The authors' responses to my comments are mostly reasonable, particularly to my request regarding the critical velocity for erosions. Following my suggestions, the authors added a table and enriched the discussions on this problem. It has led to much clearer explanation of the simulation results.

I agree with the authors that the time-dependent sediment modeling and analysis may be beyond the scope of the present study. The revised modeling approach by including the erosion models of Mill et al. (1977) represents a good alternative.

However, I am not fully convinced by the authors' response regarding the friction factor that it is not important in the present simulations. Yes, friction factor might not be important in deep water, but they may play a role in sediment transport in shallow water, especially the flow speed at the points, a (190 m), b (45 m), and d (120 m), shown and discussed in Figure S1. There are no descriptions or

discussions on the friction factor used in the model in the manuscript. So, the model did not consider the bottom friction at all?

Response to the review of our resubmitted manuscript:

Contamination of 8.2 ka cold climate records caused by the Storegga tsunami in the Nordic Seas

[Text in blue is our response, text in black is the reviewers]

We are grateful that the reviewers of our revised manuscript are satisfied with our response and changes to the original manuscript. Reviewer #1 says: “Overall the authors have responded satisfactorily to both reviewers’ comments” and reviewer #2 says: “The revised manuscript is a much-improved version from the original. Following my suggestions, the authors added a table and enriched the discussions on this problem. It has led to much clearer explanation of the simulation results. The revised modeling approach by including the erosion models of Mill et al. (1977) represents a good alternative.

However, there are still a few things and questions raised by the two reviewers that we are happy to clarify and rephrase in our 2nd revision of our manuscript.

Below we address point by point to the comments and questions by the reviewers:

Reviewer 1

(line) 31. ... ‘might have been’ , better ‘probably were’.

Thanks for pointing this out, we have changed the text accordingly.

(line) 55-8 and 223-4. It is not clear why the authors persist in citing the antique diagram of Hjulstrom (other than Nordic solidarity) which is incorrect in both its designation of the deposition field and in plotting the erosion resistance of fine sediment as a function of grain size - see work of Dade et al. Post-war (WW II) and modern treatments of critical erosion do not mention Hjulstrom, for example the books of Yalin, Raudkivi, Mehta and Nielsen.

Please note that we revised our interpretation of the erosion based on the boundary layer thickness (William and Fuhrman, 2016) in the previous revision. We replaced the grain size values from Hjulström with those from Sundborg and included the equations from Miller et al., as suggested in the previous review. Thus, we have assessed several different erosion models and not just one.

We know that the Hjulström diagram has its flaws and weaknesses, especially for the fine grain sizes, but for pedagogical reasons the diagram of Hjulström visualize the relationship between sediment erosion, transportation, and deposition in an easy understandable way. Therefore, in our revised version we keep the Hjulström reference in lines 55-58, as an introduction to the topic, but delete Hjulström from line 223-234. The latter was a mistake by us in the previous revision; we are not using Hjulström’s diagram to estimate erosion of grains, but rather Sundborg’s diagram and the Miller et al., equation (see table S2).

88. (102) I think the authors need to specify where their maximum flow velocity occurs, either at the

seafloor or is it at the top of the boundary layer (the latter).

Yes, we agree that this needs clarification. The paragraph dealing with the flow velocity is that beginning at line 88 on the previous submission. To make it clear that the simulated velocity applies only down to the top of the boundary layer, we have changed the sentence (line 88) to: “**The simulated maximum flow velocity, representing the main part of the water column, reflects the maximum wave amplitude and water depth**”. (New text underlined.) In the paragraph following line 88 it becomes clear that the simulated velocity is the top of the boundary layer.

We have also added this explanation to the figure text in line 102. Here it now says: **Fig. 1 | Maximum flow velocity of the water column down to about 1–10 m above the sea floor during the Storegga tsunami**. (New text underlined.)

And similar for figure text to Fig. 2:

Fig. 2 | Simulated tsunami surface elevation (blue line) and flow velocity down to about 1–10 m above the sea floor (dotted) during the Storegga tsunami at the location of marine sediment cores with an 8.2 ka layer. (New text underlined.)

155-7. pachyderma

Thanks for pointing out this misspelling. We have changed text accordingly.

369, if they persist, needs date.

This was a typographical error and has now been fixed. Thank you.

397, 413, 423, capitalisation,

We have checked the capitalization of these three references and, as far as we can see, they are consistent with the published titles’ bibliographical details. We will however address any issues flagged in typesetting.

Reviewer 2

However, I am not fully convinced by the authors’ response regarding the friction factor that it is not important in the present simulations. Yes, friction factor might not be important in deep water, but they may play a role in sediment transport in shallow water, especially the flow speed at the points, a (190 m), b (45 m), and d (120 m), shown and discussed in Figure S1.

We very much agree that the friction is important in interpreting the effect of the boundary layer. For this we considered the boundary layer thickness influenced by bottom friction for each sediment core site using the results from the simulations by Williams and Fuhrman (2016), see Methods section. We first computed the current velocity in the mid water column in the tsunami simulations. We then corrected for the boundary layer effect and friction when determining the effective velocity for the erosion at each core site. Bottom friction does not affect much the propagation of the first cycles of

the wave propagation (see e.g. Davies et al., 2020). To remedy, we have changed the figure text to Fig. S1:

Supplementary Fig. 1. Simulation of wave height and flow speed of the water column down to about 1-10 m above the sea floor at locations surrounding Storegga. (New text underlined.)

There are no descriptions or discussions on the friction factor used in the model in the manuscript. So, the model did not consider the bottom friction at all?

The propagation model for predicting the flow velocity throughout the main part of the water does not include friction as it is not important for the deep-water propagation. However, the friction *is* taken into account in the interpretation of the erosion at the discussed sediment core sites, through boundary layer values from Williams and Fuhrman (2016), see Methods section.

References:

Davies, G., Romano, F., and Lorito, S. (2020). Global dissipation models for simulating tsunamis at far-field coasts up to 60 hours post-earthquake: multi-site tests in Australia. *Frontiers in Earth Science*, 8, 598235.

REVIEWERS' COMMENTS

Reviewer #2 (Remarks to the Author):

It's good to know that the authors did consider friction in the modeling of each core site and added clarification in the texts. I have no more comments. My recommendation is to accept the manuscript for publication.